# The Deformation Characteristics, Fracture Behavior and Strengthening-Toughening Mechanisms of Laminated Metal Composites: A Review

**Kuan Gao, Xin Zhang \*, Baoxi Liu \*, Jining He \*, Jianhang Feng, Puguang Ji, Wei Fang and Fuxing Yin**

Research Institute for Energy Equipment Materials, TianJin key laboratory of materials laminating fabrication and interfacial controlling technology, School of Materials Science and Engineering, Hebei University of Technology, TianJin 300132, China; gaokuanhbgy@163.com (K.G.); 13933765819@126.com (J.F.); jipuguang@hebut.edu.cn (P.J.); fangwei@hebut.edu.cn (W.F.); yinfuxing@hebut.edu.cn (F.Y.)

\* Correspondence: zhang_xin@hebut.edu.cn (X.Z.); liubaoxiliubo@126.com (B.L.); hbgdjn@126.com (J.H.); Tel.: +86-022-60204057 (X.Z. & B.L.); +86-022-60202991 (J.H.)

**Abstract:** Multilayer metal composites have great application prospects in automobiles, ships, aircraft and other manufacturing industries, which reveal their superior strength, toughness, ductility, fatigue lifetime, superplasticity and formability. This paper presents the various mechanical properties, deformation characteristics and strengthening–toughening mechanisms of laminated metal matrix composites during the loading and deformation process, and that super-high mechanical properties can be obtained by adjusting the fabrication process and structure parameters. In the macroscale, the interface bonding status and layer thickness can effectively affect the fracture, impact toughness and tensile fracture elongation of laminated metal matrix composites, and the ductility and toughness cannot be fitting to the rule of mixture (ROM). However, the elastic properties, yield strength and ultimate strength basically follow the rule of mixture. In the microscale, the mechanical properties, deformation characteristics, fracture behavior and toughening mechanisms of laminated composites reveal the obvious size effect.

**Keywords:** laminated metal composites; fracture behavior; strengthening-toughening mechanism; rule of mixture; size effect

---

## 1. Introduction

Strength and toughness are important indices to evaluate the mechanical properties of metal base materials, and achieving this strengthening–toughening aim of the metal matrix composites is the unremitting pursuit goal of materials scientists [1,2]. However, due to the addition of reinforcement, metal matrix composites can obtain high mechanical strength by sacrificing their ductility and toughness. Herein, low ductility and toughness always lead to fracture failure rapidly and catastrophic accidents, which seriously limit the widespread applications of metal matrix composites [3]. About the strengthening–toughening methods of metal matrix composites, traditional investigations mainly focus on the three aspects: (1) Changing the types, compositions and size of reinforcements. In general, in situ reinforcement is better than the extra reinforcement due to the perfect interface bonding state, while nanosize reinforcement with low volume fraction can obviously improve the mechanical strength. However, these advantages can also result in the high cost and difficulty of well-distribution [4]; (2) striving for the uniform distribution of reinforcements. However, many previous works have proven that metal matrix composites with uniform distribution of reinforcement only reveal a limited strengthening effect, while the ductility and toughness are always decreased sharply [5]; (3) by

continuously reducing the internal microstructure and fault scale, in order to achieve the aim of preventing dislocation movement. That is to say, the design idea is to increase the density of point, line, face and body faults, including dispersion, dissolution, work hardening, grain refinement and phase transformation strengthening mechanisms. Grain refinement is the only way to improve mechanical strength without decreasing ductility and toughness. However, generally speaking, it is no longer suitable when the grain size less than 1 μm [6–8]. Recently, the increasingly harsh service condition and extreme environment bring forward higher requirements to the strength and toughness of light metal matrix composites. Therefore, it is necessary to develop the novel design idea of strengthening–toughening, and optimize the macroscale and microscale fracture modes on metal matrix composites. This task has also been one of the research focuses of materials [3,4].

Over the past century and a half, with the development of aerospace, automotive and shipbuilding industries, there has been an urgent demand for lightweight, high-strength and high-toughness materials. Materials scientists adjusted artificially the alloy element and chemical compositions, and regulated microstructure and substructure, and with the aid of heat treatment and hot working, a series of novel steels with high performance emerged at a historic moment [9–11]. Herein, typical high strength steels contain dual-phase (DP) steel, bainite steel, martensite steel and martensitic aging (maraging) steel [12–14], as well as high ductile steel, such as strain-induced martensite (TRIP) steel and strain-induced twinning (TWIP) steel [15,16]. Their yield strength and uniform ductility are shown in Figure 1 in detail. It is observed that super-high strength steels have low ductility and toughness, while super-high ductile steels display relatively low yield strength, which always limits their further application and industrial development [10]. Recently, researchers expect to achieve the strengthening and toughening effect by adjusting the relative proportion between hard phase (martensite and bainite) and soft phase (austenite and ferrite). However, strain always concentrates at the soft phase sites during the loading and deformation process, and the soft phase and interfaces make it easy to form voids and cracks. Premature necking and fracture failure of steels will occur with the congregate and confluence of the voids and cracks. Meanwhile, hard phase cannot bear enough force or stress due to the interface debonding phenomenon, and the strengthening effect is poor. Moreover, the composition and content between hard phases and soft phases mutually influence each other, and are hard to control accurately during the heat treatment and hot working processes, where it is very difficult to improve the strength and toughness [17]. Therefore, a novel design idea and research route are necessary to achieve the aim of strengthening –toughening.

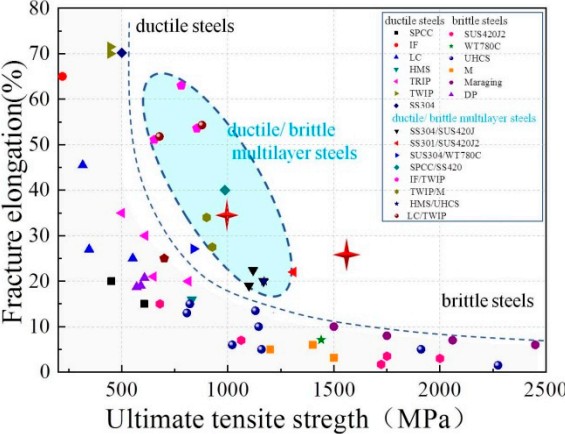

**Figure 1.** The range of tensile strength and fracture elongation of steel compared with those of other metallic materials. Adapted from [10], with permission from publisher: Elsevier, 2019.

Natural shell materials can provide reference and design ideas for achieving the strengthening –toughening aim for metal materials. As is known to all, shells are made up of $CaCO_3$ mineral with a volume fraction of 95% and protein with volume fraction of 5%, respectively. However, the fracture

energy and bending strength of shells are 3000 times and 10 times higher than that of the constitute $CaCO_3$ crystal plate, respectively. Biologists found that the super-high deformation behavior and fracture toughness of shells are mainly attributed to the abduction, delamination, bridging and interface toughening effect of the laminated structure [4,18–20]. Actually, ancient blacksmiths used two or several steels with different hardness to fabricate multilayer steels by the multiple folding and forging technique, and 1 mm thick plates contain two thousands layers. The above multilayer steels have super-high strength and toughness, which are always used in the swords weapons field, such as the Damascus sword and Samurai sword (katana) [21–23]. Kimura, Inoue and Yin et al. [24] used medium temperature groove rolling technology to fabricate multilayer nano-fibrous steels. The elongated laminated fiber texture structure and unique interface can obviously toughen the metal at the low temperature range, which is attributed to the severe delamination and crack bifurcation during the bending and impact fracture processes, and the fracture mode is similar to that of chopsticks and bamboos. It is expected to be applied to the mechanics and building field at the alpine environment.

It is revealed that the laminated metal matrix composites can improve the strength and toughness of single traditional metals. After stacking and laminating, the microstructure and properties of laminated structure materials can be independently assorted without interference along the layer direction. Therefore, the volume fraction and chemical compositions of soft and hard phases can be independently selected. When the laminated metal matrix composites are subjected to the uniaxial tensile testing, strain and stress will be evenly divided into the soft and hard layers. That is to say, the stress concentration can be effectively relieved. Therefore, the deformation compatibility and uniform plastic deformation capacity of laminated metal matrix composites can be effectively improved to a certain extent [22,25]. As shown in Figure 1, the rolled multilayer steel realizes a high strength and ductility balance; all the mechanical properties are located at the right of dotted line. Herein, rolled and aged multilayer TWIP/maraging steel reveals a superior tensile strength of 1527 MPa and total elongation of 23.5% [10]. The fabrication method and mechanical properties of multilayer steels are shown in Figure 2. Figure 2a,b shows the schematic diagram of hot roll and multiple folding–forging process of multilayer steels. Severe plastic deformation at the high temperature range during the hot rolling and hot forging process results in the strong interface bonding between soft steel and hard steel. Lesuer et al. [21] reported that multilayer steel containing high strength carbon steel (UHC) and mild steel reveals superior impact toughness and low ductile–brittle transition temperature compared to that of an individual steel layer, as shown in Figure 2c. Therefore, multilayer steel can extend the service temperature range at the alpine environment.

The present work concluded the deformation behavior and fracture mechanisms of laminated metal matrix composites based on the current research status and facing problems, and interface instability, interface delamination, superplastic deformation, and various toughening mechanisms during the fabrication and loading processes were investigated in detail. This paper also reported the effects of layer thickness, layer thickness ratio, interface bonding status, tunnel cracks on the mechanical properties and fracture behavior. Through the reasonable adjusting chemical compositions and contents of constitute layers, as well as the interface characteristics and fabrication parameters of laminated metal matrix composites, the aim of present work is in order to provide the design idea and theoretical basis of strengthening-toughening effect.

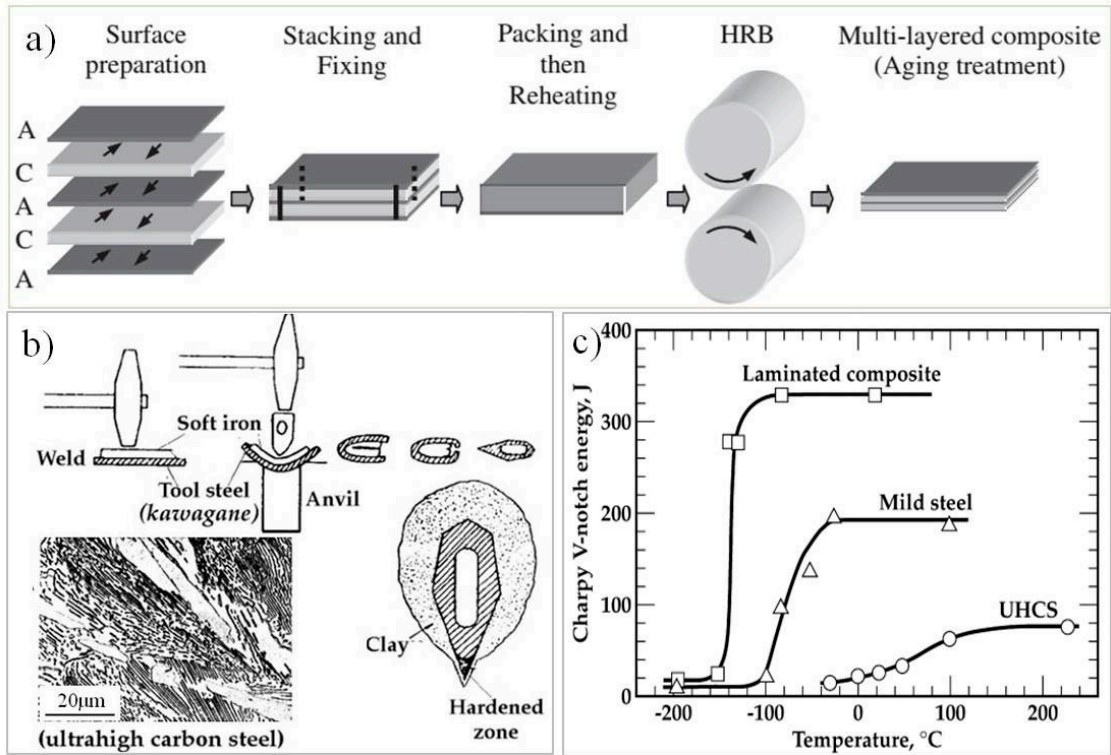

**Figure 2.** The schematic diagram of fabrication process and impact properties of multilayer steel. (**a**) hot rolling process; (**b**) multiple folding forging process; (**c**) impact toughness vs experimental temperature. Adapted from [21], with permission from publisher: Taylor and Francis, 2019.

## 2. Deformation Behavior of Laminated Metal Matrix Composites

### 2.1. Strain Localization Delaying Phenomenon

The deformation behavior of laminated metal matrix composites is different from that of traditional metal matrix composites. Traditional metal matrix composites exhibit obvious localized stress and strain concentration during the deformation process, and the interface between matrix and reinforcement is easy to debond, due to the super-high stress and strain gradients. Meanwhile, the metal matrix is easy to form premature localized necking and localized shear bands, leading to premature fracture failure of metal matrix composites [3].

However, laminated metal matrix composites can effectively delay the strain localization and stress concentration phenomena [26]. Zhang et al. [27] reported that the addition of the Al layer can obviously delay the diffuse necking phenomenon of the Cu matrix during the cold rolling process. Nanometer grain exhibits super-high strength and hardness, while it is easy to premature fracture failure due to strain localization. The coarse grain has a superior fracture elongation, while relatively low strength.

In 2011, Lu et al. [28,29] reported that the laminated gradient grain structure based on a coarse copper matrix can be successfully fabricated by surface mechanical grinding treatment (SMGT), which combines the advantages of coarsened and refined nanometer grains, revealing superior tensile strength of 400 MPa and fracture elongation of 60%. Liu et al. [26] found that the addition of the TiB$_w$/Ti composite layer effectively delayed the local necking of the Ti layer. The test shows that the fracture elongation of multilayer Ti-TiB$_w$/Ti composite is 20.5%, which is better than the whole Ti (17.5%). In the field of electronic packaging, a large number of metal films and connect wires are used in the flexible electronic products. Free metal films are easily fractured by localized necking during the tensile, bending and folding process, and bucking and splitting occur under compression and fatigue loads [30]. Li et al. [31] reported that when the film/substrate assembly affords the tensile

stress, the elastic substrate inhibits the premature localized necking behavior of metal film due to the existence of strong clad interface, resulting into high uniform deformation. Huang et al. [32] has proved that unsupported free metal films break at a relatively low strain, usually less than 1%, and form a localized necking fracture in a very small strain range. The metal film adhering to the polymer substrate can inhibit localized necking at the fracture strain range from 1% to 20%, and the corresponding schematic diagram is shown in Figure 3. The above phenomenon is due to the moderate interface bonding between polymer substrate and Cu film. The polymer substrate reveals superior uniform deformation capacity, leading to the strain localization-delaying behavior of the Cu film. However, this Cu film exhibits interface debonding behavior at high tensile strain due to the moderate interface bonding. Therefore, the localized necking occurred at the interface debonding region. The severe strain concentration and triaxial stress are also located at the necking zone, leading to grain coarsening and growth. Suo et al. and Xue et al. [33–35] simulated the three tensile deformation modes of film/substrate assembly under the plane strain state using finite element analysis based on J2 deformation theory, which are under three different types of soft substrate, medium stiff substrate and stiff substrate, respectively, and fracture modes of metal films transmit from small strain fracture, and multiple necking to a large strain uniform deformation mode. Relating to the laminated metal matrix composites with strong interface, Serror et al. [36] found that multiple neckings are localized at the small strain stage using bifurcation theory and a finite element model (FEM), and the number of multiple neckings is gradually increased with the increase of tensile strain. They are mutually competing with each other, which is similar to the uniform plastic deformation stage of traditional metallic materials. When the tensile strain reaches to the critical strain at the bifurcation point, one of multiple necking will propagate sharply, leading to the localized plastic instability. Therefore, in the macro-scale, laminated metal matrix composites reveal an obvious diffuse necking phenomenon.

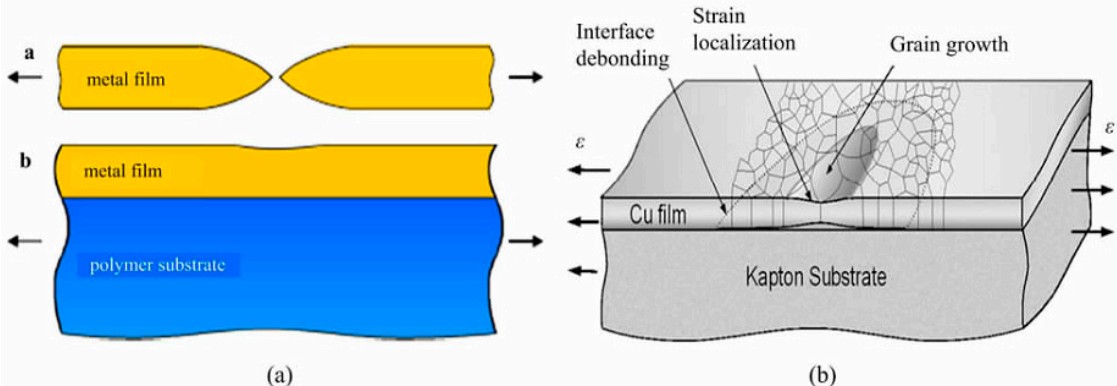

**Figure 3.** The model of tensile deformation and damage metal film: (**a**) freestanding metal film and adhere to the polymer substrate; (**b**) the mutual competing mechanism of interface debonding and strain localization. Adapted from [32], with permission from publisher: Elsevier, 2019.

Consider the strain rate is a constant, the Considere equation is expressed as follows [26,37,38]:

$$\sigma \geq \left(\frac{d\sigma}{d\varepsilon}\right) \tag{1}$$

$$\varepsilon \geq n \tag{2}$$

Herein, $\sigma$ indicates true stress, $\varepsilon$ symbolizes true strain, $\frac{d\sigma}{d\varepsilon}$ is the strain hardening rate and $n$ represents the work hardening exponent. Based on the relationship between strain and stress, the true stress is gradually increased, while the strain hardening rate is gradually decreased with the increase of true strain, when the true stress is equal to the value of strain hardening rate, and the true strain reaches to the value of the work hardening exponent. At the same time, the diffuse necking and localized shear bands will occur. Therefore, laminated metal matrix composites containing soft

layers with low necking strain and hard layers with high necking strain can obtain high uniform plastic deformation capacity.

## 2.2. Interface Instability Phenomenon

Generally speaking, the clad interface of multilayer metal composites is straight after hot rolling at a lower rolling reduction, but after hot rolling and cold rolling at a higher reduction, such as accumulative roll bonding [39–41], the clad interface will have wave shape. The raw bimetal composites containing substrate and cladding can change into partial individual substrate without surface cladding, such as stainless steel cladding, aluminum cladding and copper cladding; the cladding reveals obvious localized necking and multiple fracture failure during the cold rolling process. That is to say, the function properties containing high corrosion resistance and high electrical conductivity of bimetal composites, have disappeared [42–45]. Du and Fan et al. [46] proposed that the clad interface of the multilayered Ti/Al composites after cold rolling revealed severe multiple necking and nonuniform layer thickness. Semiation et al. [47] reported that the cold-rolled Al/Fe bimetal composites had severe interface unevenness accompanied by strain localization. Actually, the periodic multiple localized necking of cold-rolled multilayer composites can be taken as plastic bifurcation behavior proposed by Stief et al. [39]. Hutchinson et al. [38,48] reported that the strain localization at the clad interface will cause severe interface debonding and a serious decline in the overall performance of the multilayer metal composites. It is reported that when multilayer metallic composites are subjected to tensile and bending loads, a severe localized necking is presented, while at the cold rolling state, a variety of multiple necking phenomena were formed at the interface, mainly due to the inconsistent deformation behavior of different metal materials [49]. Chauhari et al. [50] found that the multiple periodic necking will become serious at high cold rolling reduction. such as Ti–Al [50], Ni–Al [51], Fe–Cu [52], Mg–Al [53], Cu–Nb [54], Ti–Ni [55], Fe–Ni [56] multilayer metal composites. Moreover, the straight-wave transition of clad interface has been investigated by a finite element model (FEM) [57]. In addition, Yu et al. [58] found that the hard metal layer will gradually form severe localized shear bands and interface delamination as the number of rolling passes increases. Liu et al. [59] reported that as the reduction ratio increased, the interface of the multilayer steel showed a beautiful wavy morphology, as shown in Figure 4. Among them, the soft Q235 steel has a uniform thickness, while the hard SUS304 shows severe localized necking and fracture failure.

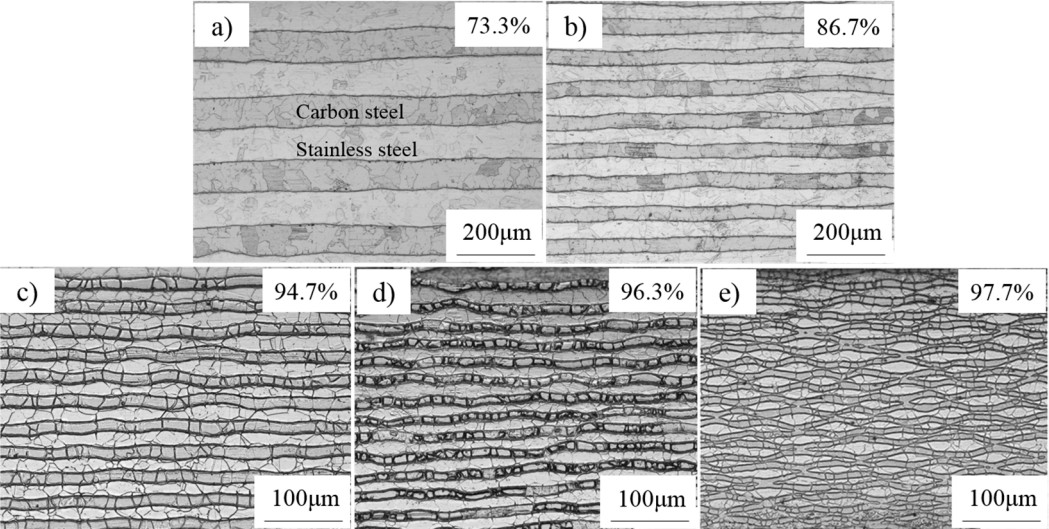

**Figure 4.** The optical microstructure (OM) of multilayer steels with different rolling reduction ratio (**a**) 73.3%; (**b**) 86.7%; (**c**) 94.7%; (**d**) 96.3%; (**e**) 97.7%. Adapted from [59], with permission from publisher: Elsevier, 2019.

Hutchinson, Stief et al. [38–40] found that interfacial instability is a kind of plastic bifurcation mode, in which homogeneous metals are uniformly deformed at the initial deformation stage and then form heterogeneous plastic deformation or local plastic deformation, which is called bifurcation mode or strain localization. Assuming the metallic materials are incompressible, strain rate independent and anisotropic, and the constitutive equation is expressed as follows:

$$\bar{\sigma}_{11} - \bar{\sigma}_{22} = 2\mu^*(\varepsilon_{11} - \varepsilon_{22}) \tag{3}$$

$$\bar{\sigma}_{12} = 2\mu\varepsilon_{12} \tag{4}$$

Herein, $\bar{\sigma}_{\alpha\beta}$ is the Jaumann derivative of Cauchy stress, $\varepsilon_{\alpha\beta}$ is the Euler strain rate, $\mu$ and $\mu^*$ are the incremental elastic and shear modulus, respectively. Assuming the flow function is expressed as follows:

$$v_1 = \frac{\partial \varphi}{\partial x_2}, \ v_2 = \frac{\partial \varphi}{\partial x_1} \tag{5}$$

The flow function must follow the following equation:

$$\left[\mu + \frac{1}{2}(\sigma_1 - \sigma_2)\right]\varphi_{,1111} + 2(2\mu^* - \mu)\varphi_{,2211} + \left[\mu - \frac{1}{2}(\sigma_1 - \sigma_2)\right]\varphi_{,2222} = 0 \tag{6}$$

Herein, $\varphi_{,1111}$ represents $\frac{\partial^4 \varphi}{\partial^4 x_1}$, $\varphi_{,2211}$ represents $\frac{\partial^4 \varphi}{\partial^2 x_1 \partial^2 x_2}$, $\varphi_{,2222}$ represents $\frac{\partial^4 \varphi}{\partial^4 x_2}$. $\sigma_1$ and $\sigma_2$ are nominal Cauchy stress, it is difficult to obtain exact solutions for such equations. In general, the critical strain values of plastic bifurcation are obtained by numerical calculation. Through simulation of the laminated metal matrix composites under cold rolling, Stief et al. found critical bifurcation strain is higher than experimental values, which indicated that the bifurcation strain depends on the rolling reduction ratio and rolling pressure. The wavy interface can be obtained by adjusting the interface slide.

*2.3. Superplastic Deformation Behavior*

Relating to the plastic deformation behavior of metallic materials, the fracture elongation depends on the strain hardening exponent (*n*) and strain rate sensitivity factor (*m*), respectively [60]. Laminated metal matrix composites revealed that the elastic properties, yield and ultimate tensile strength are approximately fitting to the rule of mixture (ROM), while the fracture elongation cannot follow the ROM. The fracture elongation of multilayer metal matrix composites should be located in the intermediate value of the constitute metals if neglected of m value. However, the actual tensile ductility of laminated metal matrix composites is always lower or higher than the constitute metals [61,62]. That is to say, the fracture elongation of multilayer metal matrix composites heavily depends on the strain rate sensitivity factor (m). Based on the Power–Hollomon equation, the flow stress of metallic materials is expressed as follows:

$$\sigma = K\left[\varepsilon^n + m\ln\left(\frac{\dot{\varepsilon}}{\dot{\varepsilon}_r}\right)\right] \tag{7}$$

Herein, $K$ is the stress constant, $\dot{\varepsilon}_r$ is the reference strain rate, high strain hardening exponent (n) represents high uniform plastic deformation capacity, and high strain rate sensitivity factor (*m*) reveals superior tensile deformation behavior [61,62]. The m value does not delay localized necking, but also retains a series of diffuse necking. Multiple necking is carried out to improve fracture elongation. Therefore, laminated metal matrix composites can achieve superior fracture elongation by adjusting and optimizing the n and m values of the constitute metallic layers.

It is well known that metal matrix composites or super-high strength metallic materials are difficult to achieve plastic formability by the traditional way at room temperature because of their high strength, hardness, low fracture elongation and work hardening capacity. In many cases, superplastic behavior can be used in plastic forming at high temperature. For example, the superplastic forming method

has already accounted for more than 50% in the titanium alloy application field [63–66]. Superplastic forming does not only save raw materials and reduce processing costs significantly, but also significantly improves the forming efficiency and makes it easy to manufacture complex parts. The Damascus knife is a multilayer steel produced by repeated forging of the high and medium carbon steel by heating up to 650 °C with superior superplastic deformation capacity [67]. Sherby et al. [68–70] reported that multilayer high carbon steel/ferrite stainless steel obtained a superior strain rate sensitivity factor of 0.3 and superplastic deformation ductility of 400% at the working temperature of 835 °C when the strain rate is $10^{-3}$ s$^{-1}$. Grishaber et al. [71] found that the relationship between stress rate and strain rate in the superplastic tensile testing is in agreement with the creep equation in the constant strain deformation model. This model can be used to predict the relationship of strain rate, tensile temperature and volume fraction of the non-superplastic phase in the laminated metal matrix composites. It is indicated that the superplastic properties of constitute layer with coarse grain can be obtained by lamination design.

Figure 5 shows the relationship between the fracture elongation of multilayer metallic composites and n, m values, and it is interesting to note that superior tensile ductility of laminated metal matrix composites can be obtained by lamination with two constitute layers possessing low fracture elongation [72,73]. For example, when a metal plate with high n and low m is laminated with a metal plate with low n and high m, laminated metal matrix composites can obtain high fracture elongation. Herein, multilayer steel with stainless steel [74] and high carbon steel, laminated Ti-TiB$_W$/Ti composites [75] by reaction hot pressing and multilayer Ti/Al composites [76] prepared by hot rolling, all fit to the above rule.

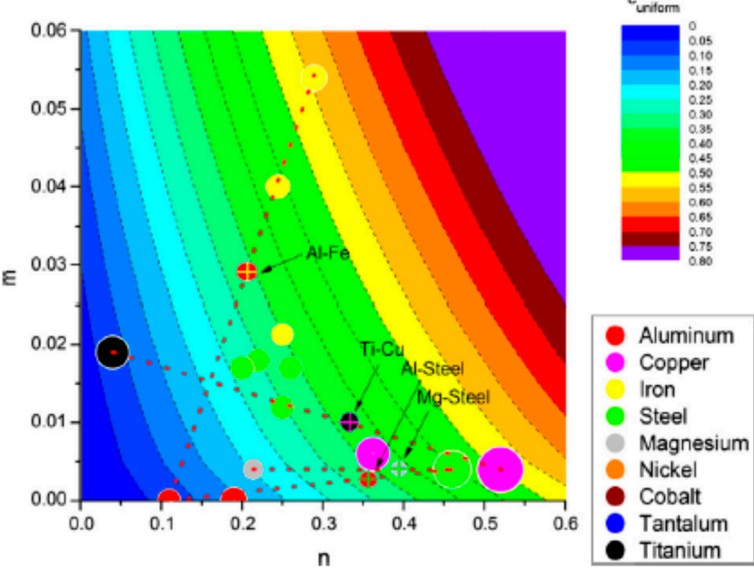

**Figure 5.** The relationship between fracture elongation of laminated metal composites and n, m of constitute layer. Adapted from [73], with permission from publisher: Elsevier, 2019.

## 2.4. Interface Delamination Behavior

Interface delamination behavior is always generated at the laminated metal matrix composites when it affords the tensile, bending and impact loading [21]. The toughening effect of interface delamination is different for the loading method. For example, in the process of tensile testing, interface delamination may make the constitute layers deformation incoordination, resulting in the decrease of overall plastic deformation capacity [77]. However, in the process of bending and impact testing, the interface delamination cracks can play an obvious toughening effect [78]. Of course, the toughening effect is also related to the length of delamination crack. Song et al. [19] reported that during the fracture process of synthetic laminated ceramics and natural laminated shells, the main crack all spreads along the interfaces. However, the crack length of the former is about four times higher than

that of the latter. According to the principle of fracture mechanics, a long interface delamination crack has great influence on the fracture resistance of materials [79].

During the bending and impact loading process, the interface delamination formation process of laminated metal matrix composites is proposed by Li et al. [80]. There are two stresses ahead of crack tip, where the stress ($\sigma_{xx}$) parallel to the direction of external force at the crack tip reveals the maximum value, while the stress ($\sigma_{yy}$) perpendicular to the external load appears the maximum value at a certain distance from the crack tip. Based on the equal strain condition, $\sigma_{yy}$ is the function of the elastic modulus of each layer, and the elastic modulus of the hard layer is higher than that of the soft layer. Therefore, the distribution of $\sigma_{yy}$ is discontinuous. In order to satisfy the interface stress equilibrium condition, the stress distribution of $\sigma_{xx}$ at the crack tip is continuous. According to the elastic mechanics, the calculated maximum stress of the $\sigma_{xx}$ is one fifth of external stress. Assuming that a weak interface is located at a certain distance of crack tip, and the interface is perpendicular to the main crack, then the main crack will propagate along the weak interface and form an interface delamination crack [25,80,81].

Liu et al. [26] investigated the profile tensile fracture characteristics and found that obvious interface delamination cracks are presented at the laminated Ti-(TiBw/Ti) composites by diffusion welding. At the initial tensile deformation stage, the deformation behavior of soft layer and hard layer is fitting to the equal strain model.

At the elastic deformation stage, the difference of elastic modulus and Poisson's ratio between soft layer and hard layer will result in interface strain and interface stress ($\sigma_{in}$), and the value of $\sigma_{in}$ is far lower than the interface bonding strength of laminated metal matrix composites. The deformation trend of the soft layer and the hard layer is gradually different with the increase of the longitudinal strain. In general, the soft layer always forms premature necking due to the rather low work hardening rate, and the necking phenomenon is more obvious with the increase of tensile strain. The transverse stress at the clad interface will continue to increase when it reaches to the interface bonding strength. Herein, the interface delamination phenomenon will occur in the laminated metal matrix composites.

In order to quantitatively explain the interface delamination crack of laminated metal matrix composites during the tensile testing process, Zhang et al. [82] provided the beam model as shown in Figure 6. The localized necking is firstly located at the soft layer, and the hard layer must bend due to strong interface. Herein, the hard layer will bend a certain deflection like a cantilever beam. In this way, the relationship between the deflection of a cantilever beam and the load per unit is as follows:

$$\delta = \frac{11P_0L_0^4}{120EI} \tag{8}$$

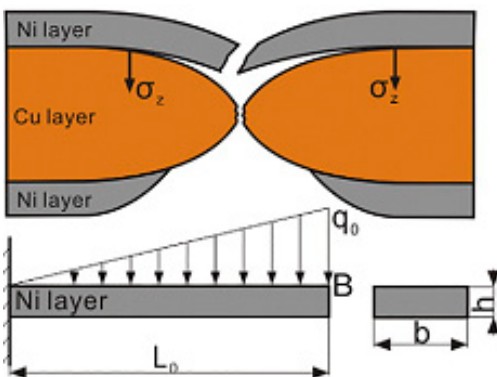

**Figure 6.** The schematic diagram of interface delamination crack evolution process. Adapted from [82], with permission from publisher: Elsevier, 2019.

Here $E$ is the elastic modulus of the hard layer, $L_0$ is the length of cantilever beam, $I$ is the inertia moment of cantilever beam.

$$I = \frac{bh^3}{12} \tag{9}$$

Herein, $b$ and $h$ are the width and thickness of hard layer. Finally, it is concluded as follows:

$$\delta = \frac{1.1\sigma_z L_0^4 r_{s/h}^3}{Et_s^3} \tag{10}$$

Herein, $r_{s/h}$ represents the layer thickness ratio between soft layer and hard layer. $t_s$ represents layer thickness of soft layer. When the laminated metal matrix composites afford same stress $\sigma_z$, the greater the thickness ratio between the soft layer and the hard layer is, the greater the deflection of the hard layer, which reveals that the interface delamination is easy to occur when it experiences sufficient necking behavior.

## 3. Toughening Mechanism and Influence Factor

### 3.1. Laminated Structure

The comprehensive mechanical properties of the laminated metal matrix composites severely depend on the composition and proportion of the constitute layer, especially the layer number, layer thickness and layer thickness ratio etc. Price et al. [83] reported that the thickness and volume fraction of soft layers play important roles in the mechanical properties and fracture behavior of laminated metal matrix composites. As a toughening purpose, the addition of the soft layer has greatly adjusted and optimized the strength, toughness and stiffness of laminated metal matrix composites. Liu et al. [84] reported that when the thickness of the hard layer is fixed, the strength is gradually increased, while the fracture elongation is decreased with the decrease of soft layer thickness, and the fracture morphologies of this soft layer has an obvious ductile–brittle transition phenomenon. Cui et al. [85,86] and Liu et al. [87] found that fixing the layer thickness of soft and hard layers, the strength is increased, while the toughness is decreased with the increase of the volume fraction of the hard layer, and the overall fracture morphologies have an obvious ductile–brittle transition. Syn et al. [88] found that the number of interface delamination cracks is decreased, while fracture elongation is sharply increased with the decrease of layer thickness, and Liu et al. also found this rule. Meanwhile, in the microscale, the number of tunnel cracks is obviously increased with the decrease of layer thickness, which plays an effectively toughening effect and delays the trend of instantaneous fracture. Therefore, the fracture elongation is enhanced in the whole [89].

So far, the size effect of multilayer metal composites remains an unsolved problem. Zhang et al. [90–95] prepared two types of multilayer metal composites with different layer thicknesses by physical vapor deposition, and in detailed studied the effects of the layer thickness and bonding strength of metal composites on the fracture and strengthening–toughening behavior. It is observed that the shear fracture failure mode at the cross interface of the multilayer Au/Cu composite using nanoindentation is tested at the micron or submicron level, while strain localization and plastic bifurcation behavior are presented at the nanometer level.

Zhang et al. [96] reported the nanoindentation of laminated Cu–Cr and Cu–Ni composites prepared by physical vapor deposition. As the layer thickness is located at the nanometer level, the strength hardness of the multilayer metal matrix composites is enhanced. However, the ductility is ambiguous. Previous research has reported that the enhancement of strength and hardness at the nanometer scale is attributed to the continuous elastic deformation initiated by the dislocation-free zone [97]. With the increase of layer thickness from nanometer scale to micron scale, the deformation and fracture behavior always exhibits the brittle–ductile transition effect.

Inoue and Nambu et al. [98,99] found that a large number of macrocracks crossing the overall layer appeared at the brittle layer, and the relatively ductile layer effectively blunts these cracks, revealing an

obvious toughening effect. However, increasing the layer thickness ratio between brittle layer and ductile layer, the thick brittle layer can cause premature fracture of the multilayer steel. Herein, the thicker the martensitic stainless steel layer is, the lower the ductility, and the mechanical properties of the laminated metal composites on the microscopic scale show a significant size effect. Studies of Gao [18] have shown that when the crack size is located at the micron scale, as shown in Figure 7, there are a severe localized stress concentration at the crack tip. However, when the crack size is located at the nanometer size, the stress intensity at the crack tip and overall sample is uniform, and the nanometer crack reveals defect insensitivity behavior, so the nanoscale brittle layer is beneficial to strengthening–toughening the multilayer composite material.

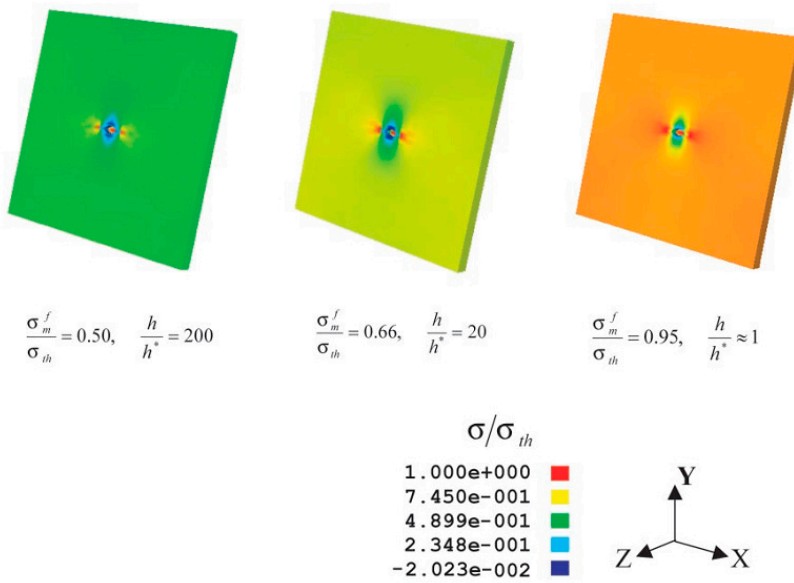

**Figure 7.** The schematic diagram of stress distribution of mineral platelet containing a microscale crack. When the *h* is near the critical thickness *h*\*, the stress concentration is relieved and the strength reaches to the theoretical strength. Adapted from [18], with permission from publisher: PANS, 2019.

### 3.2. Interface Structure

The stronger interface bonding strength can improve the deformation coordination and overall plastic deformation capacity when it is subjected to tensile loading, thus increasing the fracture elongation of laminated metal matrix composites. Strong interface bonding strength can inhibit the premature formation of strain localization and improve the hard workening capacity, and the probability of plastic bifurcation is greatly reduced. There are many ways to improve the interface strength. In this paper, the interface bonding strength is gradually increased with the increase of vacuum degree, rolling reduction ratio and rolling temperature, and the interface strength is higher than that of the carbon steel substrate [100–103]. Herein, the hot-rolled stainless steel-clad plate reveals severe carbon element diffusion, leading to the formation of a decarburized layer and carburized layer, as shown in Figure 8a, and the decarburized layer's absence of pearlite phase cannot afford high stress. Therefore, shear fracture failure is always located at the decarburized layer as shown in Figure 8b. Liu et al. [104–106] used the hot pressing to join and pin the interface by reacted TiB whisker between Ti and TiB$_2$, and the interface bonding strength is higher than that of the soft layer. The interface alloy elements' diffusion is decreased, while the fracture elongation is increased by adding the Ni interlayer. During the accumulative roll bonding process (ARB), the interface strength and toughness can be improved by adding the SiC and TiO$_2$ powders [107–109].

When the laminated metal matrix composite is subjected to the bending, impact and fatigue loads, the weak interface bonding strength can play a toughening effect. The main crack will propagate into the weak interface with the increase of stress, which causes the crack to deflect along the weak interface

due to a large number of defects at the clad interface, and even the main crack will propagate along the weak interface to form the interface delamination cracks. Therefore, the stress concentration at the crack tip will be relieved and the original triaxial stress turns into the biaxial stress or uniaxial stress state. Meanwhile, the main crack is shielded. When the force continues to increase, the main crack at the interface crack tip is deflected and penetrates into the next layer. After a multi-fracture along the interface, the crack propagation mode combines the cross layer crack and interface delamination cracks, and the crack propagation path is greatly improved. Therefore, the fracture toughness and fracture work are significantly improved [21,25].

In many laminated structure systems, most dissimilar metals may diffuse each other and generate a certain diffusion layer and reaction layer. These reacting layers are often an intermetallic, ceramic or metastable layer. When they are in the form of a separate block, it will show high brittleness, poor corrosion resistance and low strength, and while it is taken as the interlayer, it will play a toughening effect to a certain level. For example, in the system of $TiB_w$/Ti composites, it reveals a low toughness and often disastrous fracture failure. However, in the system of laminated Ti-$TiB_w$/Ti composites, a considerable number of tunnel cracks are formed during the tensile and bending process, and these tunnel cracks cannot propagate into the soft layer, so it will absorb a lot of crack propagation work and play an important toughening effect. In the research of stainless steel-clad plates, a certain thick carburized layer will be located at the interface diffusion zone due to the carbon element diffusion, which has a strong intergranular fracture and intergranular corrosion tendency [25,100]. During the tensile testing process, many tunnel cracks along the grain boundaries of the carburized layer will appear, leading to the high toughness of stainless steel clad plates [26]. Qiao et al. [110] used diffusion welding to fabricate laminated Cu/Al composites, and a variety of CuAl, $Cu_2Al$ and other compound layers are located at the interlayer interface, which are prone to brittle fracture failure during the loading process. Meanwhile, the occurrence of the crack may be located at the Al layer along the grain boundary. However, these cracks cannot be extended to the Cu layer and be inhibited by the interface, if the number and size of tunnel cracks reach to certain values, the whole laminated metal matrix composites cannot afford high load, and one of tunnel cracks further propagates into the soft layer until the final fracture failure of the whole laminated metal matrix composites.

During the fracture process of these laminated metal matrix composites, a variety of fracture mechanisms and toughening behavior will occur together. Figure 8c reveals the laminated structure and interface characteristics of laminated Ti-TiBw/Ti composites fabricated by vacuum hot pressing. Vacuum hot pressing is used to join and pin the interface by a reacted TiB whisker between Ti and $TiB_2$, and the interface bonding strength is higher than that of the soft Ti layer. In the series of laminated Ti-$TiB_w$/Ti composites, the delamination cracks and multiple tunnel cracks will occur together as shown in Figure 8d, which is beneficial to toughening the laminated composites [111]. Rohatgi et al. [112] reported that the mutually competed multiple tunnel cracks and transversal crack bifurcation in the $Ti_3Al$ layer are taken as important toughening mechanisms in the laminated $Ti_3Al$-Ti composites as shown in Figure 8e,f. Song et al. [19] investigated the nacre materials, and found that the mineral bridge connected to the interface played an obvious strengthening and toughening effect, and could effectively shorten the length of the delamination crack.

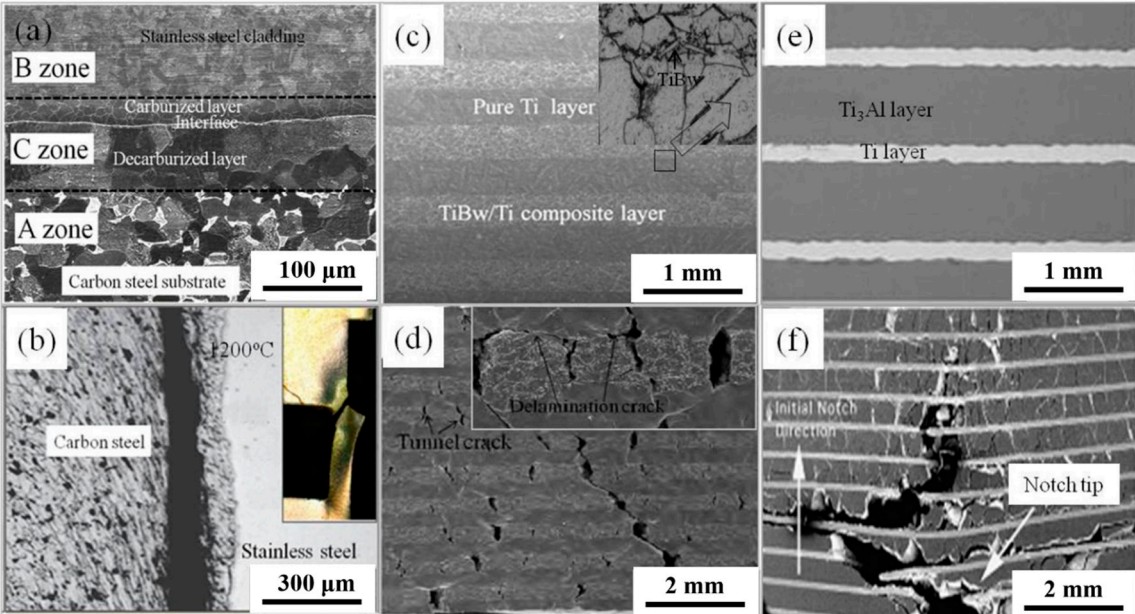

**Figure 8.** The interface microstructure and fracture characteristics of different laminated metal composites. (**a**,**b**) stainless steel clad plate, adapted from [100–103], with permission from publisher: Elsevier, 2019; (**c**,**d**) laminated Ti-TiBw/Ti composites, adapted from [104–106,111], with permission from publisher: Elsevier, 2019; (**e**,**f**) laminated Ti-Ti₃Al composites, adapted from [112], with permission from publisher: Elsevier, 2019.

### 3.3. Effect of Residual Stress

The emerged residual stress is the main strengthening–toughening mechanism of laminated metal matrix composites with strong interface during the preparation process. If the main crack propagates into the compression stress layer, the residual compression stress will reduce the tensile stress at the crack tip to a certain level, thus decreasing the stress intensity factor and crack propagation velocity. The main crack will be deflected and bifurcated in the compressive stress layer. If the main crack reaches to the tensile stress layer, a large number of micrometer intergranular cracks can be formed along the grain boundaries, which can also relieve the localized stress concentration and improve the fracture toughness of laminated metal matrix composites [110,113].

In the process of heating and cooling, the difference in thermal expansion coefficient and elastic modulus of different metal layers will lead to the high residual compression stress and residual tensile stress of laminated metal matrix composites. Zuo et al. found that the residual stress is gradually increased with the increase of fabrication temperature. The laminated metal matrix composites will form a hot crack due to increased superior tensile stress, which seriously affects the overall performance [114]. Liu et al. [100] investigated the bending fracture characteristics of laminated Ti-TiB$_w$/Ti composites and found that low diffusion welding temperature can result in the formation of weak interface and severe interface delamination as shown in Figure 9a,b, which is beneficial to improving the bending resistance of laminated composites. However, high diffusion welding temperature can effectively promote the alloy elements' diffusion behavior and form a strong interface. However, it also causes the super-high interface residual thermal stress between TiB reinforcement and the Ti matrix, leading to low fracture toughness of laminated Ti-TiB$_w$/Ti composites as shown in Figure 9c. Herein, the kind of enhancing effect is not sufficient for the strengthening and toughening matrix, which limits the practical application of laminated metal matrix composites.

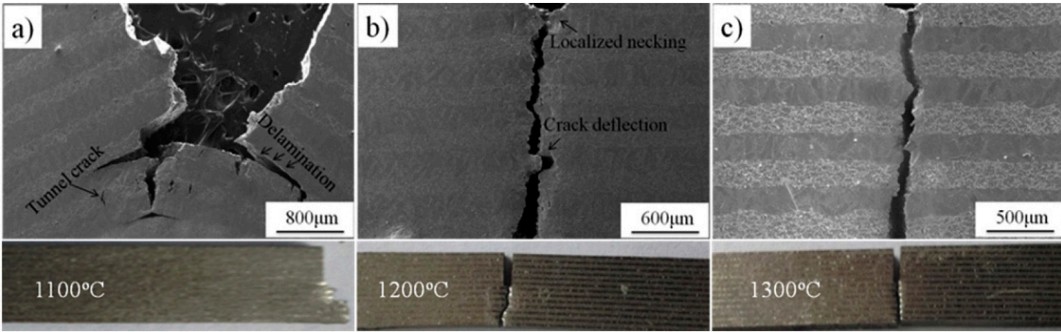

**Figure 9.** The three-point bending fracture morphologies of laminated Ti-(TiBw/Ti) composites with different diffusion welding temperatures of (**a**) 1100 °C; (**b**) 1200 °C; (**c**) 1300 °C. Adapted from [104], with permission from publisher: Elsevier, 2019.

Zhou et al. [115] investigated the laminated ZiB2-SiC ceramics and found that the residual compression and tensile stresses of individual layer can be effectively adjusted and optimized by adjusting the volume fraction of SiC, which can effectively improve the bending toughness of laminated metal matrix composites. Many investigations have shown that the residual of laminated metal matrix composites can be effectively related to the fabrication parameter, clad interface, type and volume fraction of reinforcements [111].

## 4. Conclusions

(1) In the study of laminated metal matrix composites, and the aim of strengthening and toughening can be achieved by optimizing the microstructure and interface characteristics.

(2) The fabrication parameters, laminated structure, interface characteristics and residual stress can effectively influence the comprehensive mechanical behavior of laminated metal matrix composites, and the toughening aim can be realized by adjusting the layer thickness, interface bonding status and residual compression stress reasonably.

(3) The interface instability, superplastic behavior, interface delamination and localized necking delaying effect of laminated composites can be related to the individual mechanical properties and interface microstructure, and we concluded that the layer thickness effect, residual compression stress and strong interface are beneficial to strengthening–toughening laminated metal matrix composites.

**Author Contributions:** Literature retrieval, K.G., J.F., P.J. and W.F.; writing-original draft preparation, X.Z.; writing-review and editing, B.L.; funding acquisition, J.H.; project administration, F.Y. All authors have read and agreed to the published version of the manuscript.

**Funding:** This research was funded by National Natural Science Foundation of China (No. 51601055, 51771064 and 51705129); Science and Technology Research Project of Department of Education of Hebei Province (No. BJ2017009); National Natural Science Foundation of China and Steel Research Joint Fund of Baosteel Group co., LTD. (U1860114); Technology Innovation Strategy Funding Project of Hebei Science and Technology Department and Hebei University of Technology (No. 20180106); "One Belt And One Road" science and technology innovation cooperation project of Tianjin municipality (No. 18PTZWHZ00220).

**Conflicts of Interest:** The authors declare no conflict of interest.

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
