# Peer review of "The Deformation Characteristics, Fracture Behavior and Strengthening-Toughening Mechanisms of Laminated Metal Composites: A Review"

_metals, doi:10.3390/met10010004_

Round 1
Reviewer 1 Report
The review is presenting some interesting facts and it is written in an understandable way. The largest problems are with a formal aspects of the paper:
1- Fig. 2 is a very bad quality, especially Fig.2 c). Do not simply put scanned pages to such publication. Either redo the figures with the starting date or wind some more recent publications on this well-known topic.
2- I do not agree with this statement: "Multilayer composite materials play an important role in production and life, such as 10 automobile, ship, aircraft and other manufacturing industries" - it is simply not true since laminated steels or laminated metals are not used much in cars, etc. It seems a bit exaggerated, please rewrite it.
3- The English is not so good at certain places such as "For the last one and a half centuries, with the urgent demands of aerospace, automotive and shipbuilding industries for lightweight, high strength and high toughness materials" - here the VERB is completely missing so the sentence has no meaning. Or this "hard phase is hard to bear enough force" so the hard phase is hard? what is that supposed to mean?
Improve the English considerably or sent it for professional proofreading.
4- Many claims is not supported by references, for instance, "Herein, K is the stress constant, r 235 is the reference strain rate, high strain hardening
exponent (n) represents high uniform plastic deformation capacity, and high strain rate sensitivity
factor (m) reveals superior tensile deformation behavior." Add references to appropriate places, please.
Reviewer 2 Report
See my comments in attached file

Round 2
Reviewer 2 Report
Agree with the revised manuscript